# Cell death and biomass reduction in biofilms of multidrug resistant extended spectrum β-lactamase-producing uropathogenic *Escherichia coli* isolates by 1,8-cineole

Nicolas M. Vazquez[1,2,3], Florencia Mariani[2,3], Pablo S. Torres[3,4], Silvia Moreno[1,3]*, Estela M. Galván[2,3]*

**1** Laboratorio de Farmacología de Bioactivos Vegetales, Departamento de Investigaciones Bioquímicas y Farmacéuticas, Centro de Estudios Biomédicos, Biotecnológicos, Ambientales y Diagnóstico (CEBBAD), Universidad Maimónides, Buenos Aires, Argentina, **2** Laboratorio de Patogénesis Bacteriana, Departamento de Investigaciones Bioquímicas y Farmacéuticas, Centro de Estudios Biomédicos, Biotecnológicos, Ambientales y Diagnóstico (CEBBAD), Universidad Maimónides, Buenos Aires, Argentina, **3** Consejo Nacional de Investigaciones Científicas y Técnicas (CONICET), Buenos Aires, Argentina, **4** Instituto de Ciencia y Tecnología Dr César Milstein–Fundación Pablo Casará, Buenos Aires, Argentina

* moreno.silvia@maimonides.edu (SM); galvan.estela@maimonides.edu (EMG)

**Data Availability Statement:** All files are available from the Figshare database (https://doi.org/10.6084/m9.figshare.c.5064746.v3).

## Abstract

*Escherichia coli* is the most frequent agent of urinary tract infections in humans. The emergence of uropathogenic multidrug-resistant (MDR) *E. coli* strains that produce extended spectrum β-lactamases (ESBL) has created additional problems in providing adequate treatment of urinary tract infections. We have previously reported the antimicrobial activity of 1,8-cineole, one of the main components of *Rosmarinus officinalis* volatile oil, against Gram negative bacteria during planktonic growth. Here, we evaluated the antibiofilm activity of 1,8-cineole against pre-formed mature biofilms of MDR ESBL-producing uropathogenic *E. coli* clinical strains by carrying out different technical approaches such as counting of viable cells, determination of biofilm biomass by crystal violet staining, and live/dead stain for confocal microscopy and flow cytometric analyses. The plant compound showed a concentration- and time-dependent antibiofilm activity over pre-formed biofilms. After a 1 h treatment with 1% (v/v) 1,8-cineole, a significant decrease in viable biofilm cell numbers (3-log reduction) was observed. Biofilms of antibiotic-sensitive and MDR ESBL-producing *E. coli* isolates were sensitive to 1,8-cineole exposure. The phytochemical treatment diminished the biofilm biomass by 48–65% for all four *E. coli* strain tested. Noteworthy, a significant cell death in the remaining biofilm was confirmed by confocal laser scanning microscopy after live/dead staining. In addition, the majority of the biofilm-detached cells after 1,8-cineole treatment were dead, as shown by flow cytometric assessment of live/dead-stained bacteria. Moreover, phytochemical-treated biofilms did not fully recover growth after 24 h in fresh medium. Altogether, our results support the efficacy of 1,8-cineole as a potential antimicrobial agent for the treatment of *E. coli* biofilm-associated infections.

**Funding:** This work was supported by grant number PIP 11220130100426CO from the National Research Council of Argentina (CONICET) to EMG; grant number PICT 2017-0183 from Agencia Nacional de Promocion Cientifica y Tecnologica, Argentina, to EMG; and intramural funding from Fundacion Cientifica Felipe Fiorellino, Universidad Maimonides, Argentina, to SM. The funders had no role in study design, data collection and analysis, decision to publish, or preparation of the manuscript.

**Competing interests:** The authors have declared that no competing interest exist.

## Introduction

Uropathogenic *Escherichia coli* is the most common cause of urinary tract infections, accounting for approximately 80% of infections [1]. The routine therapy of urinary tract infections is based on the use of antibiotics such as β-lactams, trimethoprim, nitrofurantoin and quinolones in many countries. Over-use and misuse of these antibiotics increase the development of resistant bacteria [2]. Particularly, the emergence of uropathogenic multidrug-resistant (MDR) *E. coli* strains that produce extended spectrum β-lactamases (ESBL) is a serious global health problem, since it can cause prolonged hospital stay, increasing morbidity, mortality, and health care costs [3]. ESBLs are a group of β-lactamase enzymes that confer resistance to third generation cephalosporins, such as ceftazidime and ceftriaxone. Resistance genes coding for β-lactamases are often located on plasmids which also harbor resistance genes for non- β-lactam antibiotics such as aminoglycosides and trimethoprim-sulfamethoxazole [4]. Therefore, ESBL producing bacteria are commonly MDR, leaving limited antibacterial options.

Uropathogenic *E. coli* forms multicellular communities known as biofilms, residing in the bladder epithelium and also on urinary catheters [5]. Bacterial biofilms are microbial communities of cells attached to a biotic or abiotic surface and embedded in a self-produced extracellular polymeric matrix [6]. Bacteria grown in biofilms are significantly more resistant to antibiotics than planktonic cells [7]. Varied mechanisms have been proposed to elucidate the high antibiotic resistance of biofilms including restricted antibiotic penetration, decreased growth rates and metabolism, and induction of cell biofilm–specific phenotypes known as persister cells [8]. The increased resistance of uropathogenic *E. coli* to antibiotics along with the bacterial ability to form biofilms cause recurrence and chronicity of urinary tract infections [5].

In the era of increasing antibiotic resistance, the search of new antimicrobial agents effective against pathogenic bacteria in their two ways of life, planktonic and biofilm stage, is a priority need in the clinical practice [9]. Volatile oils derived from aromatic and medicinal plants, such as rosemary (*Rosmarinus officinalis*), peppermint (*Mentha piperita*), thyme (*Thymus vulgaris*), fennel (*Foeniculum vulgare*), are reported to be effective against Gram-negative and Gram-positive bacteria, viruses, and fungi [10]. These plant volatile oils are complex organic metabolites with lipophilic characteristics. The specific role of individual compounds as responsible for the antimicrobial effect has not been extensively studied [11]. In a previous study, we reported one of the main constituents of rosemary volatile oil, the monoterpene 1,8-cineole (also known as eucalyptol), which exhibited a marked antibacterial activity against *E. coli* ATCC35218 strain [12]. At the minimum inhibitory concentration (MIC) [0.8% (v/v)], 1,8-cineole showed bactericidal effect on planktonic *E. coli* cells, with membrane disruption as the bactericidal mechanism identified. Other authors reported MICs of 1,8-cineole for *E. coli* strains ≥ 0.8% (v/v) [13–15]. Nevertheless, the effect of this phytochemical on *E. coli* biofilms has not been extensively explored. In particular, little is known about the effect of 1,8-cineole on bacterial viability of MDR ESBL-producing uropathogenic *E. coli* growing in biofilms.

Thus, the main goal of this study was to analyze the antibiofilm activity of 1,8-cineole against mature biofilms of MDR ESBL-producing uropathogenic *E. coli* clinical strains by evaluating its effect on biofilm biomass and cell viability.

## Materials and methods

### Bacterial strains and inoculum preparation

*Escherichia coli* strains used in this study were isolated from adult patients and are described in Table 1. *E. coli* strains named Ec AM were isolated from urinary samples collected from

**Table 1. *E. coli* strains used in this study.**

| Strain | Description | Antibiotic resistance[a] | Source |
|---|---|---|---|
| Ec ATCC25922 | Urinary isolate | None | ATCC |
| Ec AM3 | Urinary isolate (ESBL producer) | NIT, TMS, CIP, AMC, CTX, CAZ, CEF | This work |
| Ec AM4 | Urinary isolate (ESBL producer) | NIT, TMS, CIP, AMC, CTX, CAZ, CEF | This work |
| Ec AM5 | Urinary isolate (ESBL producer) | CIP, AMC, CTX, CAZ, CEF | This work |
| Ec AM6 | Urinary isolate (ESBL producer) | TMS, CIP, AMC, CTX, CAZ, CEF, GEN, AKN | This work |
| Ec AM7 | Urinary isolate (ESBL producer) | TMS, CIP, AMC, CTX, CAZ, CEF, GEN, AKN | This work |
| Ec AM8 | Urinary isolate (ESBL producer) | NIT, CIP, AMC, CTX, CAZ, CEF | This work |
| Ec AM9 | Urinary isolate (ESBL producer) | TMS, CIP, CTX, CAZ, CEF | This work |
| Ec AM10 | Urinary isolate (ESBL producer) | CIP, AMC, CTX, CAZ, CEF | This work |
| Ec AM12 | Urinary isolate (ESBL producer) | NIT, TMS, CIP, AMC, CTX, CAZ, CEF, GEN | This work |
| Ec 07 | Urinary isolate (ESBL producer) | AMP, CIP, CTX, CAZ, CEF, CEP, GEN, NAL, | [17] |

[a] AMP, ampicillin; AMC, amoxicillin- clavulanic acid; CTX, cefotaxime; CAZ, ceftazidime; FEP, cefepime; CEF, cephalothin; NIT, nitrofurantoin; TMS, trimethoprim/ sulfamethoxazole; AKN, amikacin; GEN, gentamicin; NAL, nalidixic acid; CIP, ciprofloxacin.

ATCC, American Type Culture Collection; ESBL, extended spectrum β-lactamase.

patients admitted to a medical center at Buenos Aires (Argentina) between 2017 and 2018 [16]. Strain Ec07 was isolated from a patient with polymicrobial CAUTI at Hospital Pirovano (Buenos Aires City, Argentina) [17]. Microbiological identification and antimicrobial suscepti- bility testing were carried out by standard methods. In vitro susceptibility tests were inter- preted based on CLSI breakpoints [18]. *E. coli* strains were examined for ESBL production by a double-disk synergy test using ceftazidime, cefotaxime and cefepime with and without clavula- nic acid according to CLSI guidelines [18]. *E. coli* clinical strains used in this study were iso- lated as part of routine clinical hospital procedures to diagnose infection and hence ethical approval was not required, according to the corresponding institutional guidelines.

Isolates were maintained in the laboratory as frozen stocks (at –80˚C) in Luria-Bertani (LB) broth supplemented with 15% glycerol. Inocula for assays were prepared as follows. Strains were streaked on Tryptic Soy Broth (TSB)-agar plates and grown overnight at 37˚C. Subse- quently, individual colonies were used to inoculate TSB (3 ml) and were incubated overnight at 37˚C and 200 rpm. Then, each inoculum was properly diluted in M9 minimal medium sup- plemented with 0.8% glucose in order to obtain $10^7$ cells ml$^{-1}$.

## Biofilm formation assays

Bacterial inocula in M9 supplemented with 0.8% glucose ($1 \times 10^7$ cells ml$^{-1}$) were placed in 96-well (200 µl per well) or 24-well (1 ml per well) polystyrene plates (DeltaLab, Barcelona, Spain) and incubated statically at 37˚C. Adhesion to polystyrene surface was allowed for 3 h and then the medium was replaced every 24 h for up to 3 d. At selected time points, biofilms developed in 96-well plates were washed three-times with sterile 0.9% NaCl before biomass quantification by crystal violet staining (absorbance measurement at 595 nm) [19]. All crystal violet assays were performed in technical quadruplicate and, for each plate, four wells were used as blanks containing sterile growth medium. Experiments were done in biological tripli- cates Biofilm biomass levels were classified as highly positive ($A_{595} \geq 1$), low-grade positive ($0.2 \leq A_{595} \leq 1$), or negative ($A_{595} \leq 0.2$) [20].

For quantification of cultivable cells, biofilms developed in 24-well plates were washed with sterile 0.9% NaCl before mechanical disruption from the surface as previously described [21]. The bacterial suspensions obtained were serially 10-fold diluted, plated on TSB-agar plates,

and grown for 16 h at 37˚C for enumeration of colony forming units (cfu). Experiments were done in biological triplicates and technical duplicates were performed.

## Determination of 1,8-cineole minimum inhibitory concentration

The minimum inhibitory concentration (MIC) of 1,8-cineole was determined by using the broth microdilution method, with minor modifications [12,18]. In brief, assays were performed in 96-well plates and using M9 supplemented with 0.8% glucose and containing 0.5% Tween 80 (200 µl final volume). Tween 80 (0.5%) was added to the medium to enhance phytochemical solubility [22,23]. 1,8-cineole (Sigma, MO, USA) dilutions (0.25–2%, v/v) in medium were prepared from an 80% (v/v) pure compound solution in ethanol and mixed with each bacterial strain at an initial inoculum of $1 \times 10^6$ cells ml$^{-1}$. The plates were then incubated at 37˚C for 24 h and bacterial growth assessed by measuring the absorbance at 595 nm. The MIC was defined as the 1,8-cineole concentration able to inhibit 90% of bacterial growth after 24 h incubation. As previously shown [12,22,23], bacterial growth was not affected by addition of 0.5% Tween 80 ($A_{595nm}$ 0.824 ± 0.046 *vs*. 0.839 ± 0.031 in the absence and in the presence of Tween 80, respectively). Controls containing 0.5% (v/v) ethanol, that correspond to the amount of ethanol present in the highest concentration of phytochemical tested [1,8-cineol 2% (v/v)] did not significantly inhibit bacterial growth (less than 2% inhibition). Experiments were done in biological triplicates and technical triplicates were performed.

## Biofilm susceptibility to 1,8-cineole

Mature biofilms (3 d-old) were washed with 0.9% NaCl, then, the indicated concentration of 1,8-cineole in M9 supplemented with 0.8% glucose and 0.5% Tween 80 were carefully added on top of the biofilms, and the plates were incubated statically at 37˚C. Controls (untreated) were carried out by replacing the culture medium by fresh medium. Medium supplemented with 0.5% Tween 80 was also assayed as control, giving similar result than medium without this surfactant. Vehicle controls were assessed using the ethanol concentrations corresponding to each phytochemical dilution used in medium supplemented with 0.5% Tween 80 (ethanol concentrations of 0.03, 0.06, 0.12, 0.25, 0.50%, v/v, corresponding to 1,8-cineole concentrations of 0.12, 0.25, 0.50, 1.00, 2.00%, v/v, respectively). After 15 to 180 minutes of incubation, the medium was removed, biofilms washed with 0.9% NaCl and biofilm biomass and cell viability was determined as explained before.

Assays to investigate biofilm regrowth after phytochemical treatment were performed as follow. Mature biofilms (3-d-old) developed in 24-well plates were treated with 1% 1,8-cineole (v/v) for 1 h. Vehicle controls were assessed using 0.25% ethanol. After treatment, biofilms were washed three-times with 0.9% NaCl and then fresh M9 medium supplemented with 0.8% glucose was added to the wells and plates were incubated at 37˚C. At 6 h and 24 h, the medium was removed, biofilms were washed with 0.9% NaCl, and cell viability was determined as explained before. Experiments were done in biological triplicates and technical duplicates were performed.

## Biofilm imaging

Ec AM7 biofilms were formed on 18-mm glass coverslips, as described above. Three-days-old biofilms were treated with 1% (v/v) 1,8-cineole during 1 h. Controls were carried out as explained above. Biofilms were further stained using the live/dead BacLight$^{TM}$ Bacterial Viability Kit (Thermo Fisher Scientific, Waltham, MA, USA) containing SYTO®9 green-fluorescent nucleic acid stain and the red-fluorescent nucleic acid stain, propidium iodide, which was handled following the provider's recommendations. Observation of biofilms was done using a

Carl Zeiss LSM 800 confocal laser scanning microscope (Zeiss, Oberkochen, Germany). For each biofilm, three image stacks were taken with a z-step size of 1 μm. Unstained and single-stained slices for each dye were used to monitor and subtract all respective background signals. The Zeiss ZEN Microscope Software version 3.0 was used for generation of orthogonal and 3D images. COMSTAT 2.1 (www.comstat.dk) [24,25] and the ImageJ software distribution FIJI [26] were utilized for biomass calculations and to quantify the viable (SYTO®9; green), dead (propidium iodide; red) and colocalized (SYTO®9 + propidium iodide; yellow) parts of the biofilms from the image z-stacks. Colocalized fluorescence was defined as part of propidium iodide staining, as the dye was able to penetrate the membrane. As it did not completely remove SYTO®9, it was subtracted from SYTO®9 staining.

## Evaluation of biofilm-detached cells

Ec AM7 biofilms grown for 3 d were treated with 1% (v/v) 1,8-cineole during 1 h. Controls were carried out as explained above. Bacteria in the surrounding media (~ 1 ml) were taken and centrifuged, and the pellet was resuspended in 1 ml of 0.9% NaCl. For quantification of viable cells, bacterial suspensions were serially 10-fold diluted, plated on TSB-agar plates, and grown for 16 h at 37°C for cfu enumeration. Additionally, cell viability was assessed by flow cytometry (BD FACSCanto II, Becton, Dickinson and Co., NJ, USA), using the live/dead BacLight™ Bacterial Viability Kit (Thermo Fisher Scientific, Waltham, MA, USA), as described [27]. As a control, bacterial cells were killed by incubating for 60 min at 28°C in 70% isopropanol. Flow cytometry analysis of propidium iodide and SYTO®9 co-stained bacteria was carried out using FlowJo software v10.0.7.

## Statistical analysis

Statistical significance between control and 1,8-cineole-treated samples was determined with either paired Student´s t-test (one-tailed) or the one-way analysis of variance (ANOVA) followed by Bonferroni post-hoc test using GraphPad Prism version 6 (GraphPad Software, San Diego, CA, USA). Differences were considered significant when P values were less than 0.05.

## Results

### Biofilm formation ability of multidrug-resistant ESBL-producing uropathogenic *E. coli* clinical strains

Urinary tract infections caused by multidrug-resistant (MDR) *E. coli* strains that produce extended spectrum β-lactamases (ESBL) have become an increasing health problem. An additional virulence factor reported for uropathogenic *E. coli* strains is biofilm formation [1]. Initially, biofilm formation ability of ten MDR ESBL-producing *E. coli* clinical isolates from urine patients was assessed over time (1–3 d) by determining biofilm biomass with crystal violet staining. Three isolates showed a biofilm biomass that increased over time (Fig 1), whereas seven strains were negative for biofilm production ($A_{595nm} \leq 0.198$ at d 3). At d 3, Ec AM7 was identified as highly-positive biofilm producer ($A_{595nm}$ of 2.684 ± 0.553), whereas EcAM10 and Ec07 were low-grade biofilm producers ($A_{595nm}$ of 0.448 ± 0.144 and 0.701 ± 0.109, respectively). Also the antibiotic-sensitive reference strain Ec ATCC25922 formed a substantial amount of biofilm at d 3 ($A_{595nm}$ of 2.207 ± 1.060). Large biomass variations were observed among biological replicates in the stronger biofilm-producer strains, particularly in the reference strain Ec ATCC25922. In this regard, many variables could affect biofilm production in microtiter plates as well as crystal violet assessment including slight variations in incubation times, incubation temperatures, little variations in the dye solution, and/or stochastic

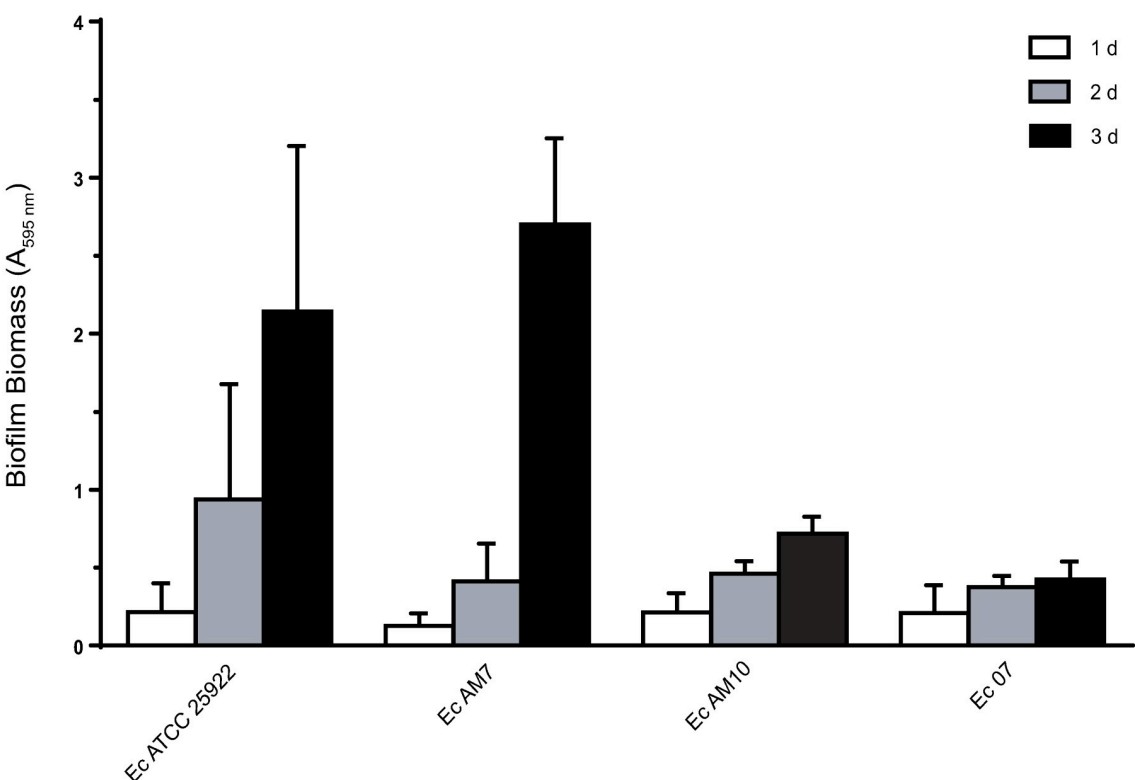

**Fig 1. Biofilm formation ability of *E. coli* clinical strains.** Biofilms were developed in M9 medium onto polystyrene plates and biofilm biomass was determined by crystal violet staining ($A_{595nm}$) after 1, 2 or 3 days. Values are means of at least three biological replicates, and error bars indicate standard deviations.

variations during the washing steps [28–30].Thus, depending on the type of biofilm and the strength of adherence that is present, some level of overestimation or underestimation of biofilm biomass might occur.

Nevertheless, based on the results obtained, the four strains selected to carry out further studies can be considered biofilm producers.

## Minimum inhibitory concentration of 1,8-cineole

The minimum inhibitory concentration (MIC) of 1,8-cineole against the selected biofilm-producing *E. coli* strains were determined (Table 2). The MDR ESBL-producing strain Ec AM10 and the antibiotic-sensitive strain Ec ATCC25922 showed MIC values in the range of 0.5–2% (v/v) 1,8-cineole. A higher MIC was observed for the MDR ESBL-producing strain Ec 07

**Table 2. Minimum inhibitory concentration of 1,8-cineole against selected biofilm-producing *E. coli* strains.**

| Strain | 1,8-cineole MIC range (%, v/v) |
|---|---|
| Ec ATCC25922 | 0.5–1 |
| Ec AM7 | >2[a] |
| Ec AM10 | 1–2 |
| Ec 07 | ≥2 |

MIC, minimum inhibitory concentration. Values from biological triplicate experiments are shown.

[a] A 14% growth inhibition was reached with 2% (v/v) of the phytochemical.

(MIC ≥2% of the phytochemical). The third MDR ESBL-producing strain under study, Ec AM7, was less susceptible to the phytochemical, showing only 14% inhibition of bacterial growth when 2% (v/v) 1,8-cineole was assayed.

## Concentration-response and time-course effect of 1,8-cineole over cell viability in pre-formed biofilms

The ability of 1,8-cineole to affect cell viability in mature biofilms was analyzed. For this purpose, the strong biofilm-producer strain Ec AM7, that generates a substantial biofilm biomass at d 3, was chosen to determine the optimal treatment conditions. First, 3-d-old biofilms were challenged with increasing phytochemical concentrations (0.125 to 2%, v/v) or the corresponding amount of its vehicle ethanol (0.03 to 0.5%, v/v) during 1 h and viable cell counts were determined (Fig 2A and 2B). Fig 2A showed that the number of viable cells in biofilms was not modified by any of the ethanol concentrations tested. However, increasing concentrations of 1,8-cineole showed a concentration-dependent detrimental effect on bacterial viability (Fig 2B). A phytochemical concentration of 0.5% (v/v) caused a 1.5-log decrease in viable cell counts in the attached biofilm, whereas both 1 and 2% (v/v) 1,8-cineole showed the highest detrimental effect in biofilm viability (a 3-log decrease of viable cells). Time-course experiments in which biofilms were exposed to 1% (v/v) 1,8-cineole evidenced the highest cell viability reduction when the phytochemical was applied for 1 h, and no higher effect was observed 3 h after treatment (Fig 2C). Altogether, these results demonstrated the efficacy of a treatment with 1% (v/v) 1,8-cineole during 1 h to significantly diminish the number of viable cells in pre-formed *E. coli* biofilms.

## Effect of 1,8-cineole over cell viability in biofilms of various MDR ESBL-producing *E. coli* clinical isolates

Next, the effect of this phytochemical treatment on cell viability in mature biofilms formed by the other *E. coli* strains under study was assayed (Fig 3). Significant reductions in the number of viable cells, ranging from 3- to 4-log, were observed in biofilms of all three tested bacteria, either sensitive to antibiotics or MDR ESBL-producers, compared to vehicle-treated controls. It should be noted that all tested strains were affected by 1% (v/v) 1,8-cineole in biofilms, independently of their susceptibility to the phytochemical when in planktonic state (MICs ranging from 0.5 to >2%, v/v). Altogether, these results clearly evidenced the antibiofilm activity of 1,8-cineole against pre-formed biofilms produced by both antibiotic-sensitive and MDR ESBL-producing strains of *E. coli*.

## Evaluation of biofilm biomass disruption by 1,8-cineole treatment

The decrease in the number of viable cells in the biofilm observed after 1,8-cineole treatment could be caused by a disruption of the biofilm structure. To investigate this possibility, biofilm biomass was determined by crystal violet staining after phytochemical treatment (Table 3). Certain variations were observed in the remaining biomass detected between independent assays; these differences can be attributed to some mechanical disruption of biofilms produced during washing steps. Nevertheless, a clear reduction in biofilm biomass was observed in all four *E. coli* strain tested after 1 h treatment with 1% (v/v) 1,8-cineole, compared to their corresponding vehicle-treated controls (48–65% decrease in biofilm biomass). This result showed that, regardless of the amount of biofilm biomass produced by each strain, this compound was able to disrupt the biofilms in a similar percentage (more than 50% of biomass reduction in all tested strains).

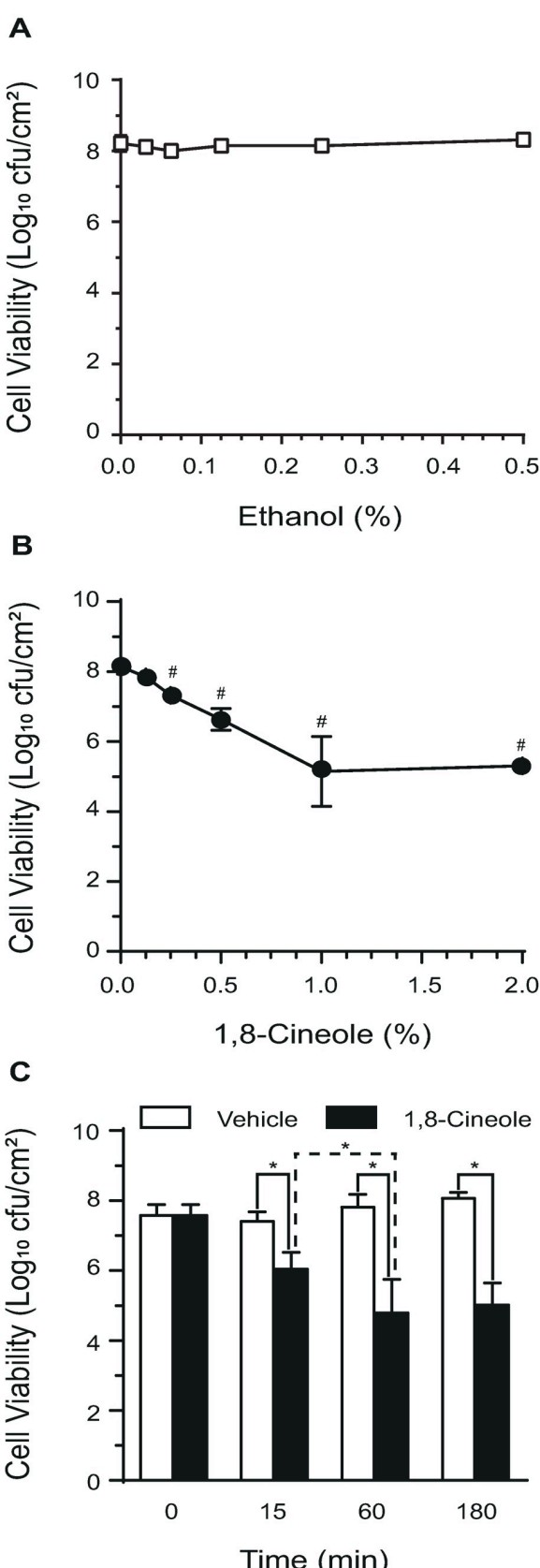

**Fig 2. Concentration-response and time-course effect of 1,8-cineole over cell viability in pre-formed *E. coli* biofilms.** Biofilms of the *E. coli* strain Ec AM7 developed for 3 d were challenged with increasing concentrations of the vehicle ethanol (A) or 1,8-cineole (B) for 1 h, and then the number of viable cells per cm² were assessed after mechanically recover cells from polystyrene plates. In (B), values are means of five biological replicates. (#) p<0.05 compared to untreated control by one-way ANOVA followed by Bonferroni post-hoc test. (C) Three-days-old Ec AM7 biofilms were exposed to 1% (v/v) 1,8-cineole or the corresponding vehicle concentration (ethanol 0.25%, v/v) at different times, and the number of viable cells assessed as explained above. Values are means of four biological replicates, and error bars indicate standard deviations. (*) p<0.05 by Student´s t test.

## Assessment of bacterial viability of surface-attached cells after exposure to 1,8-cineole by confocal microscopic analysis

To evaluate whether 1,8-cineole is capable of killing *E. coli* Ec AM7 cells in biofilms, the biofilm samples were live/dead-stained for analysis by confocal laser scanning microscopy (CLSM). Fig 4A–4C presents the representative confocal images of the studies groups. Visualization of the biofilm structure in control (without any treatment) and vehicle-treated *E. coli* biofilms showed that the majority of cells were alive and only a few dispersed dead bacteria were observed (Fig 4A and 4B). In contrast, 1,8-cineole treated biofilms evidenced mostly dead cells and, remarkably, these dead bacteria were distributed throughout the biofilm structure (Fig 4C). Quantification of live and dead biomass by COMSTSAT quantitative analysis of confocal images indicated around 83% of viable bacteria in control biofilms, whereas 95% of cells in 1,8-cineole-treated biofilm were dead (Fig 4D).

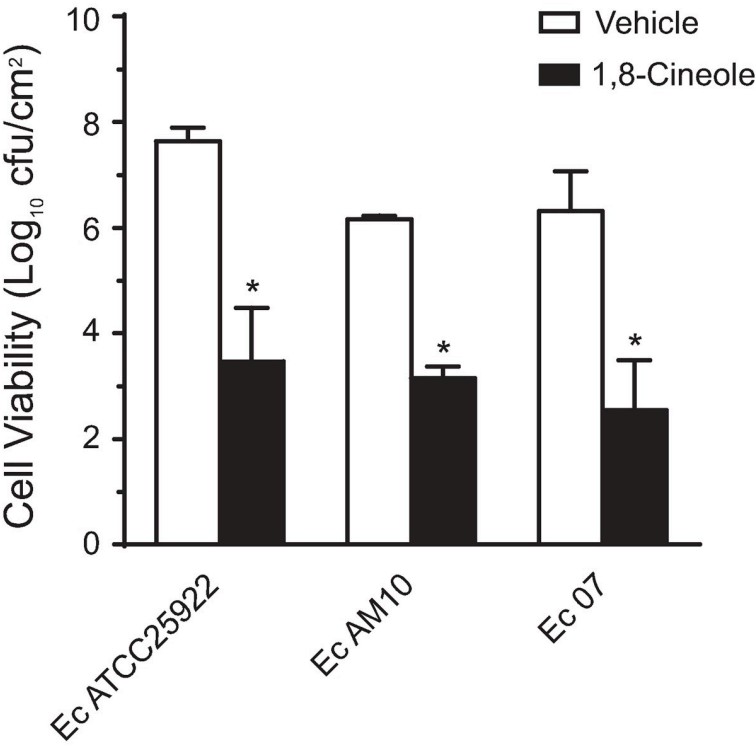

**Fig 3. Effect of 1,8-cineole over cell viability of pre-formed *E. coli* biofilms.** Three-days-old biofilms of the *E. coli* strains Ec ATCC25922, Ec AM10, and Ec 07 were challenged with 1% (v/v) 1,8-cineole for 1 h, and then the number of viable cells per cm² were assessed as described in legend of Fig 2. Values are means of three biological replicates, and error bars indicate standard deviations. (*) p<0.05 compared to vehicle by Student´s t test.

**Table 3. Effect of 1,8-cineole on pre-formed *E. coli* biofilms.**

| Strain | Biofilm biomass ($A_{595nm}$)[a] | | Biofilm disruption (%) (mean ± SD) |
|---|---|---|---|
| | Vehicle control (mean ± SD) | Treated with 1,8-C (mean ± SD) | |
| Ec ATCC25922 | 1.76 ± 0.76 | 0.74 ± 0.44 | 60 ± 12 [b] |
| Ec AM7 | 2.66 ± 0.33 | 1.34 ± 0.37 | 48 ± 18 [b] |
| Ec AM10 | 0.73 ± 0.20 | 0.39 ± 0.25 | 49 ± 21 [b] |
| Ec 07 | 0.45 ± 0.12 | 0.16 ± 0.05 | 65 ± 02 [b] |

[a] 3-d-old biofilms treated for 1 h with 0.25% (v/v) ethanol (vehicle control) or 1% (v/v) 1,8-cineole (1,8-C). Biological quadruplicates were performed.

[b] Significant difference (p < 0.05) compared to the vehicle control by Student's t-test.

These results evidenced that biofilm treatment with 1,8-cineole during 1 h produced a high level of cell death within the biofilm.

## Viability evaluation of biofilm-detached cells after 1,8-cineole treatment

As already shown here, 1,8-cineole treatment produced a significant loss of biofilm biomass, and consequent release of detached cells into the surrounding medium. It has been postulated that a good antibiofilm agent should not only attack bacteria into the biofilm but also display an action against biofilm-released cells [31]. To analyze this issue, viability of detached cells was assessed by two experimental approaches.

First, determination of cfu counting was performed (Fig 5A). In both untreated (medium alone) and vehicle-treated biofilms a substantial amount of viable bacteria were detected in the surrounding media ($7.21 \times 10^7$ and $5.39 \times 10^7$ cfu/ml, respectively). This is likely due to the reported active dispersion of cells from mature biofilms [32], considering that a minimal planktonic growth would occur in this minimal medium in 1 h. On the other hand, the number of viable cells detached from phytochemical-treated biofilms was substantially lower ($5.21 \times 10^3$ cfu/ml).

Second, detached cells were live/dead-stained for flow cytometry analysis (Fig 5B). As expected, a small proportion of cells released from vehicle-treated biofilms showed propidium iodide fluorescence signal (2.1% cells stained). Conversely, 1,8-cineole treatment caused a clear increase in propidium iodide fluorescence of the biofilm-detached cells (98.3% cells stained). This result indicated that the majority of the *E. coli* cells removed from the biofilm after the phytochemical treatment have their membrane integrity compromised.

Taken together, these results evidenced that after 1,8-cineole treatment, bacteria detached from *E. coli* biofilms were mostly dead cells.

## Evaluation of biofilm regrowth after 1,8-cineole treatment

To investigate whether the biofilm cells surviving the 1,8-cineole treatment can grow to the level of before treatment, fresh medium was added to treated-biofilms and cell viability was assessed after 6 h and 24 h at 37°C (Table 4). Viable cells in control biofilms treated with vehicle were in the range of $2.10 \times 10^7$ to $5.90 \times 10^7$ cfu/cm$^2$ in the time-period assayed. As observed earlier, 1 h exposure to 1% (v/v) 1,8-cineole (0 h post-treatment) diminished cell viability 3.5-log (to $5.80 \times 10^3$ cfu/cm$^2$). After 6 h and 24 h in fresh medium, viable cell counts of phytochemical-treated biofilms increased to $3.70 \times 10^4$ and $1.60 \times 10^5$ cfu/cm$^2$. These regrown biofilms showed at least 2.5-log lower cell counts than vehicle-treated biofilms.

The presented results evidenced the effectiveness of the 1,8-cineole treatment to limit biofilm regrowth.

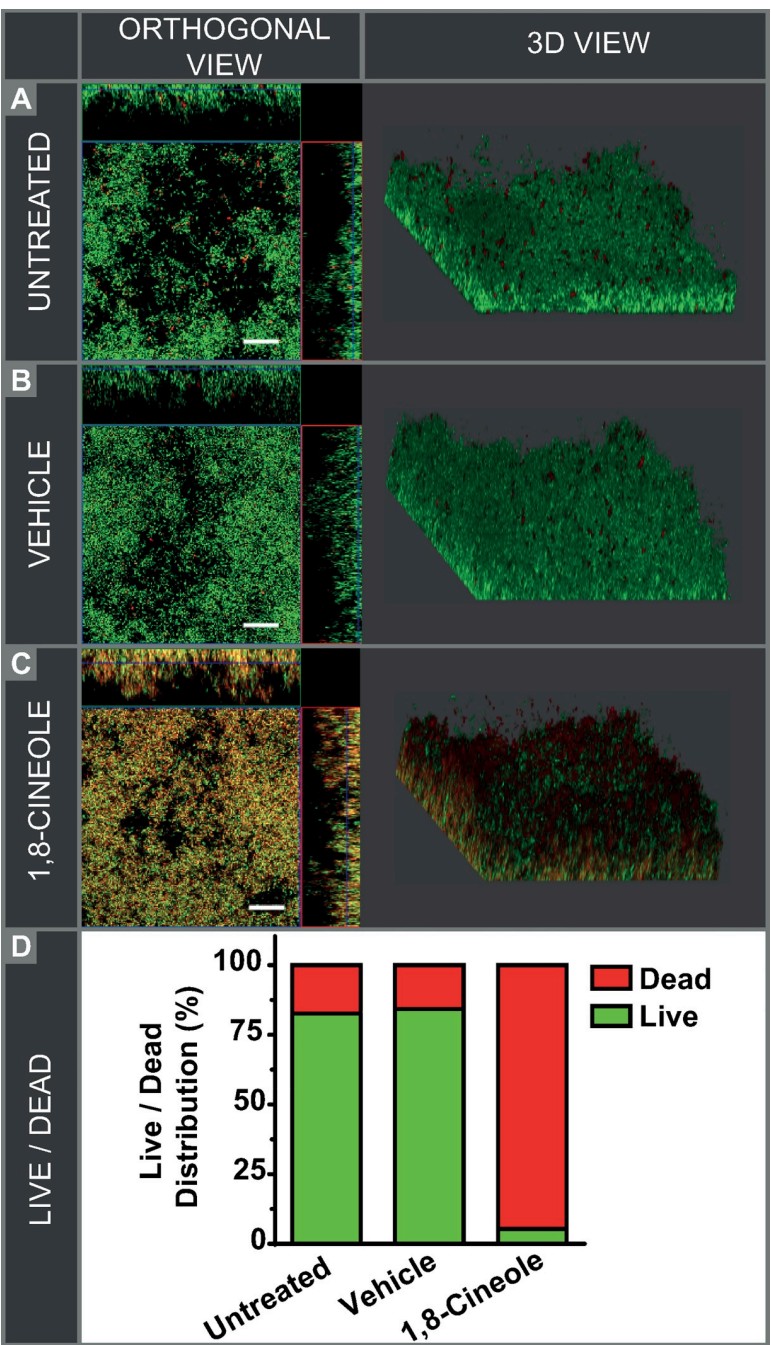

**Fig 4. Confocal laser scanning microscopy of LIVE/DEAD-stained *E. coli* biofilms.** Pre-formed biofilms (3 d-old) of the MDR ESBL-producing strain Ec AM7 were incubated for 1 h with (A) M9 medium (untreated), (B) 0.25% ethanol (vehicle), or (C) 1% (v/v) 1,8-cineole and were further incubated with the Live/Dead viability stain to show live (green) or dead (red/yellow) bacterial cells. Scale bars: 20 μm. (D) COMSTAT analysis of biomass. For each condition, the % of live and dead bacteria was calculated.

## Discussion

The extended biofilm recalcitrance toward antibiotic treatment has generated an urgent need for novel strategies against biofilm-associated infections [5]. We have previously reported that 1,8-cineole exhibits bactericidal activity against planktonic *E. coli* cells [12]. Therefore, we

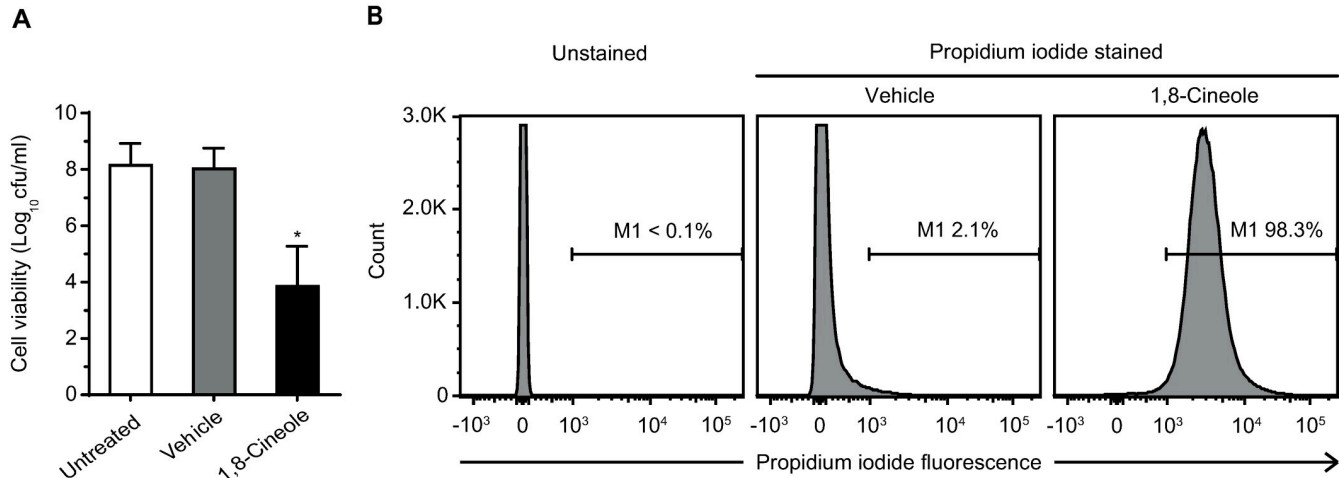

**Fig 5. Viability of biofilm-detached cells after 1,8-cineole treatment.** Biofilm-detached cells from pre-formed biofilms (3 d-old) of the MDR ESBL-producing strain Ec AM7 were collected after 1 h incubation with medium alone (untreated), 0.25% ethanol (vehicle) or 1% (v/v) 1,8-cineole. **(A)** Determination of viable cells by cfu counting. Values are means of three biological replicates, and error bars indicate standard deviations. (*) $p < 0.01$ compared to vehicle by Student´s t test. **(B)** Flow cytometry analysis after Live/Dead staining. Data were displayed as flow cytometric histograms of counted bacterial events (y-axis) associated cell fluorescence (x-axis). Marker M1 is the region that the damaged cells were stained by propidium iodide. For each sample, $10^5$ cells were analyzed.

analyzed here the antibiofilm activity of this phytochemical against MDR ESBL-producing uropathogenic *E. coli* strains that are biofilm producers.

In the present study, the incidence of biofilm formation in MDR ESBL-producing uro-pathogenic *E. coli* was 30% (n = 10). Published studies have reported a great variability in bio-film-production ability by urine-associated *E. coli* strains, ranging for 13 to 69% of total strains studied (n = 100–250) [33–35]. These variations can be explained by intrinsic differences among individual *E. coli* isolates as well as variations in the experimental conditions used to assess biofilm formation. In this regard, it has been reported stronger biofilm formation in minimal media, such as M9, than in rich media by clinical strains of *E. coli* [36]. Here, we used M9 medium supplemented with glucose (0.8%) and the static model of biofilm formation on microtiter plate for biofilm formation assays. Even though the incidence of biofilm formation we found was moderated (30%), under a clinical point of view this result is relevant because biofilm-producing MDR bacteria not only increase the chronicity of urinary tract infection but also make the infection more recalcitrant to antibiotic treatment [5].Our new findings demonstrate that 1,8-cineole diminished the total number of viable cells in mature biofilms of a MDR ESBL-producing strain, in a concentration- and time-dependent manner. A bacteri-cidal effect in biofilms was observed (viable cell reduction of 3–4 log) by applying during 1 h a

**Table 4. Regrowth of *E. coli* biofilms after 1,8-cineole treatment.**

| | Cell viability after treatment (Log$_{10}$ cfu/cm$^2$)$^a$ | | |
|---|---|---|---|
| | **0 h post-treatment** | **6 h post-treatment** | **24 h post-treatment** |
| **Vehicle control** | 7.317 ± 0.042 | 7.316 ± 0.042 | 7.789 ± 0.109 |
| **Treated with 1,8-C** | 3.765 ± 0.273$^b$ | 4.567 ± 0.064$^b$ | 5.207 ± 0.039$^b$ |

$^a$ 3-d-old biofilms treated for 1 h with 0.25% (v/v) ethanol (vehicle control) or 1% (v/v) 1,8-cineole (1,8-C) and then incubated in fresh M9 medium. Data correspond to mean ± SD of three biological replicates.

$^b$ Significant difference ($p < 0.05$) compared to the corresponding vehicle control by Student´s t-test.

concentration of 1% (v/v) 1,8-cineole (corresponding to a sub-MIC level). Mature biofilms formed by all *E. coli* strains tested in this study, both antibiotic-sensitive and MDR ESBL-producers, were susceptible to the phytochemical. Thus, the antibiofilm efficacy of 1,8-cineole reported here supports its use against *E. coli* strains forming relatively high biofilm biomasses.

As stated in the Introduction, the focus of this study was on the antibiofilm activity of 1,8-cineole against mature biofilms. At this stage, cells within the biofilm might be under stress from depleting nutrients and oxygen, and therefore this circumstance could impact the phytochemical's efficacy. In this regard, other researchers evidenced that the monoterpene carvacrol, a phytochemical with reported antimicrobial activity, was more biocidal during early biofilm development compared to mature biofilms formed by *Staphylococcus aureus* and *Salmonella enterica* [37]. Future work needs to be done to better understand the antibiofilm effect of 1,8-cineole over *E. coli* biofilms, particularly at an early developmental stage.

The observed decrease in the number of viable biofilm-forming cells after 1,8-cineole treatment could be attributed either to a disruption of the biofilm structure or to a direct bactericidal effect in the biofilm. Concerning the first possibility, the compound was able to decrease the biomass of pre-formed biofilms by approximately 50%. Moreover, the majority of the detached cells were found dead by flow cytometric analysis. Regarding the second hypothesis, we visualized by confocal microscopy that most of the adherent cells remaining in biofilms after 1,8-cineole challenge were dead, as judged by uptake of the normally membrane-impermeant dye propidium iodide.

The regrowth of biofilms after antimicrobial treatments has been considered as a critical reason for the persistent biofilm infection [38]. Our findings indicate that biofilm regrowth after 1,8-cineole treatment is limited, as 24 h after treatment there was still a 2.5 logs lower cell counts than in control biofilms.

From a clinical point of view, monotherapy have limited efficacy in the treatment of urinary tract infections caused by MDR ESBL-producing *E. coli* and combination of antimicrobial agents may be of clinical interest [39]. In this regard, as 1,8-cineole killed the majority of, but not all, *E. coli* cells forming the biofilm, further investigation focused on possible synergistic interactions of this phytochemical with common antibiotics would have important clinical implications for the treatment of biofilm-related infections involving *E. coli*.

The antimicrobial activities of individual compounds that are main constituents of plant volatile oils have been extensively studied in planktonic bacteria, however, relatively few of them have been investigated against biofilms formed by uropathogenic *E. coli*. Phytochemicals such as cinnamaldehyde from cinnamon oil [40], carvacrol from oregano oil [41], and thymol from thyme red oil [41,42] have been reported to reduce *E. coli* biofilm formation at sub-MIC concentrations. Nevertheless, none of these studies analyzed whether the compounds were able to disrupt pre-established biofilms since they were added at the beginning of the experiment.

A number of studies have reported both MIC and bactericidal effect of 1,8-cineole against planktonic *E. coli* to be in the range of 0.25–6.25% (v/v) [12,13,15,43,44]. However, there are few reports in the literature where 1,8-cineole has been tested as an anti-biofilm agent against *E. coli*. In [13] the authors studied the antimicrobial efficacy of this phytochemical against an antibiotic-sensitive *E. coli* strain. In this report, a bactericidal effect on a pre-established biofilm (24 h-old) was observed after 24 h exposure to 256 g/l (27.8% v/v) 1,8-cineole, concentration corresponding to 4-times the MIC (MIC = 64 g/l, 6.25% v/v). Recently, it has been reported a 50% biomass reduction of a preformed (18 h-old) *E. coli* biofilm by applying a sub-MIC concentration of 1,8-cineole during 24 h [45]. However, no information regarding cell viability in those treated biofilms was provided.

Our work here clearly shows that 1 h challenge with 1,8-cineole caused both biomass reduction and cell death in pre-formed biofilms of MDR ESBL-producing uropathogenic *E. coli*

isolates, at concentrations that were not lethal for planktonic cells. Thus, we obtained an effective anti-biofilm effect on mature *E. coli* biofilms by applying a sub-MIC phytochemical concentration for a short period of time (1h). The reasons behind the discrepancy between our findings and other laboratory´s results are not yet fully understood. Differences in the bacterial strains used, the experimental conditions for biofilm development (type and size of the surface, culture medium, biofilm age), and treatment duration might impact the final outcome.

Although the exact mechanism behind the antibiofilm effect of 1,8-cineole against uropathogenic *E. coli* is not yet entirely known, our findings demonstrate that the phytochemical is able to partially disrupt the biofilm, as well as to directly kill bacteria within the biofilm. Notably, our confocal images revealed that the plant compound affected the entire biofilm, including not only the outermost layer but also the innermost cells of the biofilm. This behaviour is in accordance with the idea that small non-polar components of plant volatile oils, having a superior diffusion coefficient than common antibiotics, present a high biofilm penetration potential [46]. The antibiofilm activity of 1,8-cineole can be attributed, at least in part, to membrane permeabilization of biofilm-forming cells upon penetration into the biofilm structure, as this monoterpene exhibit a bactericidal activity against planktonic *E. coli* cells that is associated with injury to the cell membrane [12]. Other authors have reported *Staphylococcus aureus* biofilm inhibition by 1,8-cineole, in the context of a chronic rhinosinusitis model, that was correlated with a decrease of proliferation and a down-regulation of major key players in biofilm generation (agrA, SarA and $\sigma^B$ genes) [45].

The management of urinary tract infections has become more difficult because of the increasing prevalence of MDR strains and the inability of antibiotics to fully eradicate biofilm-embedded bacteria. Altogether, our findings suggest that 1,8-cineole exhibit an outstanding advantage in terms of agent accessibility to biofilm-based infectious diseases, overcoming tolerance antimicrobial mechanisms and causing cell death and biomass reduction in biofilms formed by MDR ESBL-producing uropathogenic *E. coli* strains.

## Conclusions

This study is the first to demonstrate the antibiofilm activity of 1,8-cineole against MDR ESBL-producing *E. coli*. Notably, the compound is able to cause substantial bacterial dead into the biofilm-attached and biofilm-released cells. Therefore, we propose this phytochemical as a potential compound in development of novel *E. coli* antibiofilm agents.

## Acknowledgments

We are grateful to Claudia Garbasz, Head of the Microbiology Service at the Hospital General de Agudos "Dr. I. Pirovano" (Buenos Aires city, Argentina) for providing the clinical bacterial isolates. We are also thankful to Dr. Marisa Gomez and Dr. Camila Ledo for help with flow cytometry experiments. NMV and FM are doctoral fellows and PST, SM and EMG are researcher members of CONICET.

## Author Contributions

**Conceptualization:** Nicolas M. Vazquez, Silvia Moreno, Estela M. Galván.

**Formal analysis:** Nicolas M. Vazquez, Pablo S. Torres, Silvia Moreno, Estela M. Galván.

**Funding acquisition:** Silvia Moreno, Estela M. Galván.

**Investigation:** Silvia Moreno, Estela M. Galván.

**Methodology:** Nicolas M. Vazquez, Florencia Mariani, Pablo S. Torres.

**Project administration:** Silvia Moreno, Estela M. Galván.

**Resources:** Pablo S. Torres, Silvia Moreno, Estela M. Galván.

**Supervision:** Silvia Moreno, Estela M. Galván.

**Validation:** Estela M. Galván.

**Visualization:** Nicolas M. Vazquez, Florencia Mariani.

**Writing – original draft:** Estela M. Galván.

**Writing – review & editing:** Silvia Moreno, Estela M. Galván.

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
