## [Decision Letter · Decision Letter 0]

1 Apr 2020

PONE-D-20-04063

Cell death and biomass reduction in biofilms of multidrug resistant extended spectrum β-lactamase-producing uropathogenic Escherichia coli isolates by 1,8-cineole

PLOS ONE

Dear Dr Galvan,

Thank you for submitting your manuscript to PLOS ONE. After careful consideration, we feel that it has merit but does not fully meet PLOS ONE’s publication criteria as it currently stands. Therefore, we invite you to submit a revised version of the manuscript that addresses the points raised during the review process.

We would appreciate receiving your revised manuscript by May 16 2020 11:59PM. To enhance the reproducibility of your results, we recommend that if applicable you deposit your laboratory protocols in protocols.io, where a protocol can be assigned its own identifier (DOI) such that it can be cited independently in the future. For instructions see: http://journals.plos.org/plosone/s/submission-guidelines#loc-laboratory-protocols

We look forward to receiving your revised manuscript.

Kind regards,

Monica Cartelle Gestal, PhD

Academic Editor

PLOS ONE

Journal Requirements:

Please ensure that your manuscript meets PLOS ONE's style requirements, including those for file naming. The PLOS ONE style templates can be found at http://www.plosone.org/attachments/PLOSOne_formatting_sample_main_body.pdf and http://www.plosone.org/attachments/PLOSOne_formatting_sample_title_authors_affiliations.pdf

Reviewers' comments:

Reviewer's Responses to Questions

**Comments to the Author**

1. Is the manuscript technically sound, and do the data support the conclusions?

Reviewer #1: Partly

Reviewer #2: Partly

2. Has the statistical analysis been performed appropriately and rigorously? 

Reviewer #1: Yes

Reviewer #2: No

3. Have the authors made all data underlying the findings in their manuscript fully available?

Reviewer #1: Yes

Reviewer #2: No

4. Is the manuscript presented in an intelligible fashion and written in standard English?

Reviewer #1: Yes

Reviewer #2: No

5. Review Comments to the Author

Reviewer #1: Vazquez and colleagues present a manuscript that highlights an important issue - antibiotic resistance. With growing resistance to currently available therapeutics - natural compounds are of growing interest as potential biofilm inhibitors.

The manuscript is well written - however I have some concerns about the methodology used and the associated explanation for the chosen methods.

- Why did the authors choose a 3 day biofilm assay? Response to antimicrobial strategies in bacterial populations can be observed at the transcript level (RNA) after ~30 minutes treatment, by growing biofilms for 3 days - my concern at this small scale is that cells within the biofilm are under stress from depleting nutrients and oxygen, and therefore will impact any live/dead assessment being made for the inhibitor.

A time course with treatment would be a good way to test how the inhibitor's efficacy is impacted by depleting biofilm health.

-Were flow experiments only performed on dispersed cells? These are likely to be live as they are active - flow should be performed on biofilm cells also to calculate quantity of live/dead. This was not clear in the manuscript.

- 1,8 - cineole may be acting as a dispersal agent - what are the potential health consequences of this in a patient? This should be discussed by the authors. Also additional dispersal assays would strengthen the manuscript. Is motility involved?

Reviewer #2: This paper's major issue is that the English is not of sufficient quality to review. I believe that the experiments conducted and the methods used are likely sufficient to address the question but at this time the paper must be largely re-written. Due to this I have been unable to fully review the paper but this was my review of the parts I could review.

I dislike the term essential oil in a scientific paper. It does not convey a clear description of what the product is. Clearer terms such as plant oil. It can also be misappropriated by alternative medicine proponents. The authors are I assume not working under the guise that the aroma of the oil is in anyway contributing to the antibacterial nature of the oil. If that hold true then the use of ‘essential oil’ should be avoided.

In terms of scientific questions, the authors layout the following approach. Using the plant extracted oil a substantial reduction in viable cells was observed (a 3-fold log reduction). This was result was obtained in both antibiotic resistant and sensitive cells. This is then expanded to say that there is a reduction in biomass. The remaining biomass was then subjected to confocal analysis of live/dead staining and showed part of the remaining adhered biomass was actually dead. Cell no longer adhered we found to be mostly dead by flow cytometry analysis.

The use of statistics seems reudementary but without access to the raw data (as required by PLOS one) it is hard to determine. I would argue that given your hypothesis and previous work a two tailed test is not correct, you are working under the hypothesis that the plant extract will reduce the biofilm and or cell viability.

Regards

James Gurney

6. PLOS authors have the option to publish the peer review history of their article (what does this mean?). If published, this will include your full peer review and any attached files.

Reviewer #1: No

Reviewer #2: Yes: James Gurney

---

## [Author Response · Author response to Decision Letter 0]

6 May 2020

May 06, 2020

Dear Monica Cartelle Gestal, PhD

Academic Editor

PLOS ONE

Please find enclosed our revised version of the manuscript number PONE-D-20-04063 entitled “Cell death and biomass reduction in biofilms of multidrug resistant extended spectrum β-lactamase-producing uropathogenic Escherichia coli isolates by 1,8-cineole” for your consideration. It has been revised and changes have been made for each specific comment of the reviewers, as addressed below (modifications are indicated, referring to line numbers in the Marked Up Manuscript file).

Reviewer #1

General comment: Vazquez and colleagues present a manuscript that highlights an important issue - antibiotic resistance. With growing resistance to currently available therapeutics - natural compounds are of growing interest as potential biofilm inhibitors.

The manuscript is well written - however I have some concerns about the methodology used and the associated explanation for the chosen methods.

Author response to general comment: We now better explain the rational of the methodology used (see L248-251; L316-318; L324-327; L340-349; L353-354).

Specific comment #1: Why did the authors choose a 3 day biofilm assay? Response to antimicrobial strategies in bacterial populations can be observed at the transcript level (RNA) after ~30 minutes treatment, by growing biofilms for 3 days - my concern at this small scale is that cells within the biofilm are under stress from depleting nutrients and oxygen, and therefore will impact any live/dead assessment being made for the inhibitor.

A time course with treatment would be a good way to test how the inhibitor's efficacy is impacted by depleting biofilm health.

Author response #1: We now better explain in the text the reason for choosing a 3 day-old biofilm (to assess the phytochemical antibiofilm activity over a mature robust biofilm with a substantial biomass, as showed by crystal violet assay) (see L248-250 and L278). 

We agree with the reviewer´s comment regarding that cells within 3-d-old biofilms could be under stress for depleting nutrients and oxygen. We now discuss the possibility that the inhibitor´s efficacy could be impacted by this situation and, therefore, that future work is needed to analyze this possibility (see L400-403, L406-408). In addition, results reported by other group, that applied the phytochemical carvacrol at different stages of biofilm development, has been included in the Discussion (see L403-406). These authors found that the compound was more biocidal during early biofilm development compared to mature biofilms.

Specific comment #2: Were flow experiments only performed on dispersed cells? These are likely to be live as they are active - flow should be performed on biofilm cells also to calculate quantity of live/dead. This was not clear in the manuscript.

Author response #2: We now better explain the rational for the experiments to analyse cell viability after 1,8-c treatment on cells detached from the biofilms (by cfu counting and live/dead staining followed by flow cytometry) (see L339-363). On the other hand, we now re-write the paragraph corresponding to assessment of bacterial viability of surface-attached cells by confocal microscopy, to clearly explain that quantification of live/dead cells into biofilms was performed by confocal microscopy followed by COMSTSAT quantitative analysis. (see L316-327).

Specific comment #3- 1,8 - cineole may be acting as a dispersal agent - what are the potential health consequences of this in a patient? This should be discussed by the authors. Also additional dispersal assays would strengthen the manuscript. Is motility involved?

Author response #3: We now better explain the effect of 1,8-cineole as partially disrupting the biofilm and causing detachment of mainly dead cells (see L340-364). Nevertheless, because we detected a minor amount of detached cells still alive, we now discuss the importance to further investigate the possible synergistic interactions of 1,8-cineole with common antibiotics to make the phytochemical treatment more efficient in a patient (see L424-430). Besides, we believe that further exploration of the biofilm detachment events caused by the phytochemical is beyond the scope of the present work.

Reviewer #2

General comments: This paper's major issue is that the English is not of sufficient quality to review. I believe that the experiments conducted and the methods used are likely sufficient to address the question but at this time the paper must be largely re-written. Due to this I have been unable to fully review the paper but this was my review of the parts I could review.

Author response to general comment: We revised the whole manuscript for proper English usage, several parts of the text has been re-written and we consider that its quality has now been improved.

Specific comment #1: I dislike the term essential oil in a scientific paper. It does not convey a clear description of what the product is. Clearer terms such as plant oil. It can also be misappropriated by alternative medicine proponents. The authors are I assume not working under the guise that the aroma of the oil is in anyway contributing to the antibacterial nature of the oil. If that hold true then the use of ‘essential oil’ should be avoided.

Author response #1: As suggested, the term “essential oil” has been changed by “plant volatile oil” or “volatile oil” all along the manuscript ( see L28,83,87,90,431,453).

Specific comment #2: In terms of scientific questions, the authors layout the following approach. Using the plant extracted oil a substantial reduction in viable cells was observed (a 3-fold log reduction). This was result was obtained in both antibiotic resistant and sensitive cells. This is then expanded to say that there is a reduction in biomass. The remaining biomass was then subjected to confocal analysis of live/dead staining and showed part of the remaining adhered biomass was actually dead. Cell no longer adhered we found to be mostly dead by flow cytometry analysis.

The use of statistics seems reudementary but without access to the raw data (as required by PLOS one) it is hard to determine. I would argue that given your hypothesis and previous work a two tailed test is not correct, you are working under the hypothesis that the plant extract will reduce the biofilm and or cell viability.

Author response #2: As the reviewer requested, we now provide access to all the raw data through the Figshare repository (https://doi.org/10.6084/m9.figshare.c.4964399.v1). According with the reviewer´s comment, we revised the statistics used and now a one-tailed Student´s t-test was applied to our data (see L202 and test results showed in the repository).

We thank the reviewers and editors for their constructive comments and suggestions. We believe that the revised manuscript is now acceptable for publication in PlosOne.

Sincerely yours,

Estela Galván

---

## [Decision Letter · Decision Letter 1]

9 Jun 2020

PONE-D-20-04063R1

Cell death and biomass reduction in biofilms of multidrug resistant extended spectrum β-lactamase-producing uropathogenic Escherichia coli isolates by 1,8-cineole

PLOS ONE

Dear Dr. Galvan,

Thank you for submitting your manuscript to PLOS ONE. After careful consideration, we feel that it has merit but does not fully meet PLOS ONE’s publication criteria as it currently stands. Therefore, we invite you to submit a revised version of the manuscript that addresses the points raised during the review process.

We look forward to receiving your revised manuscript.

Kind regards,

Monica Cartelle Gestal, PhD

Academic Editor

PLOS ONE

Reviewers' comments:

Reviewer's Responses to Questions

**Comments to the Author**

1. If the authors have adequately addressed your comments raised in a previous round of review and you feel that this manuscript is now acceptable for publication, you may indicate that here to bypass the “Comments to the Author” section, enter your conflict of interest statement in the “Confidential to Editor” section, and submit your "Accept" recommendation.

Reviewer #2: All comments have been addressed

Reviewer #3: (No Response)

2. Is the manuscript technically sound, and do the data support the conclusions?

Reviewer #2: Partly

Reviewer #3: Yes

3. Has the statistical analysis been performed appropriately and rigorously? 

Reviewer #2: Yes

Reviewer #3: Yes

4. Have the authors made all data underlying the findings in their manuscript fully available?

Reviewer #2: Yes

Reviewer #3: Yes

5. Is the manuscript presented in an intelligible fashion and written in standard English?

Reviewer #2: No

Reviewer #3: Yes

6. Review Comments to the Author

Reviewer #2: Second review of the proposed plos one article

Cell death and biomass reduction in biofilms of multidrug resistant extended spectrum β-lactamase-producing uropathogenic Escherichia coli isolates by 1,8-cineole

The authors have made several key improvements to the English as evident by the tracked changes version, however there are a number of flaws still remaining that will need addressing. Plos One does not copy edit so they must be fixed before acceptance.

The improved English has allowed me to complete a full review. I have 4 major concerns and a number of minor concerns; I will include the errors in English under the minor concerns where I have spotted them, I will not correct them only point out where I believe there is a flaw.

Major concerns

How much of the reduction is due to the presence of tween 80? Previous work using the same compound did not make use of the surfactant. Why was it used and why was it not controlled for? On line 219 the authors make mention of reduced bacterial growth. They should present this data. Did they do a tween negative control? If not, they should. Line 389-392: has to be expanded significantly, currently other published work shows a much higher level for both MIC and bactericidal of the oil. What rational do the authors have for this discrepancy?

Given that the majority of clinical isolates did not make biofilms the authors must address the rational for using a biofilm disrupting treatment. I see three possibilities, and all should be discussed. Either the medium (M9) is not sufficient to recapitulate the clinical environment, reducing the enthusiasm for this work. Or the presence of biofilm forming strains in actual clinical samples is limited raising question about the usefulness of this treatment. Third, perhaps samples collected from clinical are bias for not biofilm forming as they would presumably be easier to collect. I would ask the author to at least address point 1 and 2 and remove all work related to the strain that did not produce biofilms.

The authors use 3 different methods of approximating biofilms. Crystal violet, CFUs, and confocal live dead staining. All 3 methods gave wildly different levels of reduction. For example, why did the CFU have 3-4 log reduction while the confocal had around a 1-2 log fold? Why don’t these methods agree? This I find further concerning as the MIC does not agree with the biofilm data. I would revise the section from line 364-379 to address this problem.

Section starting on line 309. Are the cells actually released from the biofilm or are they just growing planktonically? If they are released than according to fig 5A the ethanol treatment is better at releasing cells. If they are just growing planktonically this section needs to be revised.

Minor

Line 35: English

Line 38: English

Line 63: Needs a citation

Line 68: needs a citation

Line 74: why differentiate between fungi and yeast?

Line 83-84 I would remove the QS section. It adds nothing.

Line 116: English in two instances. Freshly streaked doesn’t mean anything and you do not streak *in* agar but *on* it.

Line 117: what is the volume used for the overnights?

Line 136: Why did you only use duplicates here? This data is not robust for statistically analysis. Please state all replicate numbers in figure legends

Line 146: space between tween 80 is missing.

Line 149: state what the vehicle is and how much was used.

Line 152: was only 1 assay done as the sentence implies? Or should it be biofilm biomass and cell viability?

Line 155: English

Line 182: English

Line 216: table say equal or greater to, but text suggests it is sensitive to 2%

Line 236: English

Line 248: and other parts, the authors say at least 3 replicates, are there uneven tests? What did the authors do to reduce the P-hacking of sampling at different rates per treatment? Or is this a case of an English mistake?

Line 254-255: Why does biomass (Fig 1) not correlate with figure 3?

Line 260: English

Line 282: What is the replicate number for data in table 3

Line 288 English

Line 316 & 318: give exact numbers.

Line 366: English

Line 411: English

Reviewer #3: General comments. Vazquez and colleagues present a manuscript evaluating the antibiofilm activity of 1,8-cineole against pre-formed mature biofilms of uropathogenic multidrug-resistant E. coli clinical strains.

The manuscript is well written and experiments are presented on a rational base.

However I have the following concerns:

Specific comment #1. The authors evaluate biofilm formation of ten E. coli isolates. Only one of them presented a substantial biofilm formation and two a mild formation. Raw data presented supports these conclusions. However data dispersion of biofilm formation in biofilm-forming strains is unusual. Authors should discuss these anomalies.

Specific comment #2. Authors should analyze the low frequency of biofilm-forming isolates in the context of evaluating a possible biofilm inhibition treatment.

Specific comment #3. Fig2A. The authors show cell viability on biofilms when they increased 1,8 cineole concentration. In raw data authors present five assays for treatment experiment but only one for control. Although differences between treatment and control with concentrations > 0.5% are important, authors must demonstrated this with a test. I don´t think the curve representation is the best suitable for these result.

Authors can consider present results in two parts: ethanol in extracts is not detrimental for survival and on the other side, dependence of viability to 1,8 cineole concentration compared to non 1,8 cineole (first column on raw data).

Specific comment #4. Table 3. The dispersion values between assays are noteworthy. Although differences are significant authors should address why they have such differences between assays. Dispersion was not observed in raw data for fig 1. I assume that some disruption of biofilm was produced during washes during treatment.

Specific comment #5. Authors described viability of detached cells arguing “It has been postulated that a good antibiofilm agent should not only attack bacteria into the biofilm but also display an action against biofilm-released cells”. The results is in concordance with high rate of dead cells on biofilm. It would be more interesting and will significantly improve the impact of the work to investigate if a post-treatment biofilm is able to growth.

Specific comment #6. Authors did not demonstrate that high biofilm biomasses yielded by strains are consequence of thicker extracellular matrix. Statement in L388-L393 should be modified.

7. PLOS authors have the option to publish the peer review history of their article (what does this mean?). If published, this will include your full peer review and any attached files.

Reviewer #2: Yes: James Gurney

Reviewer #3: No

---

## [Author Response · Author response to Decision Letter 1]

20 Jul 2020

Estela M. Galvan, Ph D

Lab. of Bacterial Pathogenesis

Centro de Estudios Biomedicos, Biotecnologicos,

Ambientales y Diagnostico (CEBBAD)

Universidad Maimonides

Hidalgo 775

C1405BWE-Buenos Aires, Argentina

July 20, 2020

Dear Monica Cartelle Gestal, PhD

Academic Editor

PLOS ONE

Please find enclosed our revised version of the manuscript number PONE-D-20-04063R1 entitled “Cell death and biomass reduction in biofilms of multidrug resistant extended spectrum β-lactamase-producing uropathogenic Escherichia coli isolates by 1,8-cineole” for your consideration. It has been revised and changes have been made for each specific comment of the reviewers, as addressed below (modifications are indicated, referring to line numbers in the Marked Up Manuscript file).

Reviewer #2: Second review of the proposed plos one article

General comment: The authors have made several key improvements to the English as evident by the tracked changes version, however there are a number of flaws still remaining that will need addressing. Plos One does not copy edit so they must be fixed before acceptance. 

Author response to general comment: We addressed all marked flaws. We consider that now the whole manuscript has proper English usage.

Major concern #1: How much of the reduction is due to the presence of tween 80? Previous work using the same compound did not make use of the surfactant. Why was it used and why was it not controlled for? 

On line 219 the authors make mention of reduced bacterial growth. They should present this data. Did they do a tween negative control? If not, they should. 

Line 389-392: has to be expanded significantly, currently other published work shows a much higher level for both MIC and bactericidal of the oil. What rational do the authors have for this discrepancy?

Author response #1: Tween 80, as well as DMSO, are commonly used to increase solubility of lipophilic molecules, such as essential oils and their individual compounds, in aqueous solutions (Man and Markham, 1998; Ojeda-Sana et al., 2013; Kwiatkowski et al., 2019). The effect of Tween 80 (0.5 %) was tested, and it did not show any detrimental effect neither on planktonic growth (MIC determinations) nor on biofilms (biomass, cell viability, cell detachment). We have now rewritten Material and methods section to clarify this issue (see L 143-154 and L164-170).

Regarding the reduction of bacterial growth mentioned on line 219, we now presented the data in Table 2 (see L253). As explained above, tween negative controls have been performed. 

As requested, additional published work regarding MIC and bactericidal effect of 1,8-cineole has been included and discussed in the context of our findings (see L457-479).

Major concern #2: Given that the majority of clinical isolates did not make biofilms the authors must address the rational for using a biofilm disrupting treatment. I see three possibilities, and all should be discussed. Either the medium (M9) is not sufficient to recapitulate the clinical environment, reducing the enthusiasm for this work. Or the presence of biofilm forming strains in actual clinical samples is limited raising question about the usefulness of this treatment. Third, perhaps samples collected from clinical are bias for not biofilm forming as they would presumably be easier to collect. I would ask the author to at least address point 1 and 2 and remove all work related to the strain that did not produce biofilms.

Author response #2: We addressed this reviewer’s concern by adding a paragraph in the Discussion section (see L395-407). Additionally, strains that did not produce biofilm were removed from Fig 1.

Major concern #3- The authors use 3 different methods of approximating biofilms. Crystal violet, CFUs, and confocal live dead staining. All 3 methods gave wildly different levels of reduction. For example, why did the CFU have 3-4 log reduction while the confocal had around a 1-2 log fold? Why don’t these methods agree? This I find further concerning as the MIC does not agree with the biofilm data. I would revise the section from line 364-379 to address this problem.

Author response #3: For Ec AM7 biofilms, our results showed 2-4 log cell viability reduction by CFU counts and ⁓ 2 log reduction by confocal live/dead microscopy. These results are within the inter-experimental variation observed. In addition, while CFU counting assesses viable cells in the whole biofilm sample, confocal microscopy explores a limited number of microscopic fields of view.

Regarding the reviewer´s concern that MIC does not agree with the biofilm data, we consider that this is an important finding of the present work. Notably, the phytochemical showed better antimicrobial effect on biofilms compared to planktonic cells. This is in agreement with a recent report (Schurmman et al, 2019). We now better discuss this issue in the manuscript (see L465-475).

Major concern #4- Section starting on line 309. Are the cells actually released from the biofilm or are they just growing planktonically? If they are released than according to fig 5A the ethanol treatment is better at releasing cells. If they are just growing planktonically this section needs to be revised.

Author response #4: To address this reviewer´s comment we now included additional experimental data in Fig 5A. We also added a paragraph in the Results section regarding this issue (see L360-364 and L380).

Minor concerns

#1. Line 35: English. Response: Modified (see L35).

#2. Line 38: English. Response: Modified (see L39).

#3. Line 63: Needs a citation. Response: As requested, a citation has now been included (see L63).

#4. Line 68: needs a citation. Response: As requested, a citation has now been included (see L68).

#5. Line 74: why differentiate between fungi and yeast? Response: We regret this mistake and now the sentence was re-written (see L74).

#6. Line 83-84 I would remove the QS section. It adds nothing. Response: As requested, this sentence has been removed (see L83-84).

#7. Line 116: English in two instances. Freshly streaked doesn’t mean anything and you do not streak *in* agar but *on* it. Response: This sentence has been now modified (see L116).

#8. Line 117: what is the volume used for the overnights? Response: The requested information has now been added (see L117).

#9. Line 136: Why did you only use duplicates here? This data is not robust for statistically analysis. Please state all replicate numbers in figure legends. Response: Experiments were done in biological triplicate and technical duplicates were done. We now clarified this information in Materials and Methods section (see L129-131, L138-139). This data is robust for statistical analysis, as detailed in the raw data. All biological replicate numbers are stated in figure legends.

#10. Line 146: space between tween 80 is missing. Response: Corrected (see L162).

#11. Line 149: state what the vehicle is and how much was used. Response: Modified as requested (see L164-170).

#12. Line 152: was only 1 assay done as the sentence implies? Or should it be biofilm biomass and cell viability? Response: This sentence has been corrected (see L171).

#13. Line 155: English. Response: Modified (see L175).

#14. Line 182: English. Response: Modified (see L202-203).

#15. Line 216: table say equal or greater to, but text suggests it is sensitive to 2%. Response: For clarity, the corresponding sentence was re-written (see L241-245).

#16. Line 236: English. Response: Modified (see L270-274).

#17. Line 248: and other parts, the authors say at least 3 replicates, are there uneven tests? What did the authors do to reduce the P-hacking of sampling at different rates per treatment? Or is this a case of an English mistake? Response: We apologize for any English mistake. We now re-write these sentences to better explain what has been done (see L283, L288-289, L307-308, L327, L383).

#18. Line 254-255: Why does biomass (Fig 1) not correlate with figure 3? Response: Fig 1 shows biofilm biomass assessed by crystal violet whereas fig 3 shows cell viability of biofilms assessed by colony forming units (CFU). Whereas the crystal violet dye detects negatively-charged molecules in the biofilm (such as polysaccharides in the biofilm matrix, and lipopolisaccharides from bacterial cells), the CFU counting is the standard technique to detect viable cells. Because each technique detects different characteristics of the biofilm, no strict correlation between them is expected. Anyhow, for example, biofilm biomass and CFU numbers were higher for Ec ATCC25922 biofilms than for Ec AM10 biofilms.

#19. Line 260: English. Response: Modified (see L301-302).

#20. Line 282: What is the replicate number for data in table 3. Response: The requested information has now been added (see L327).

#21. Line 288 English. Response: Modified (see L332-333).

#22. Line 316 & 318: give exact numbers. Response: Modified as requested (see L362).

#23. Line 366: English. Response: Modified (see L430).

#24. Line 411: English. Response: Modified (see L502).

Reviewer #3

Specific comment #1: The authors evaluate biofilm formation of ten E. coli isolates. Only one of them presented a substantial biofilm formation and two a mild formation. Raw data presented supports these conclusions. However, data dispersion of biofilm formation in biofilm-forming strains is unusual. Authors should discuss these anomalies.

Author response #1: As requested, we now discuss this issue in the manuscript (see L227-230).

Specific comment #2: Authors should analyze the low frequency of biofilm-forming isolates in the context of

evaluating a possible biofilm inhibition treatment.

Author response #2: As requested, we now discuss this findings (see L396-408).

Specific comment #3: Fig2A. The authors show cell viability on biofilms when they increased 1,8 cineole

concentration. In raw data authors present five assays for treatment experiment but only one for control. Although differences between treatment and control with concentrations > 0.5% are important, authors must demonstrate this with a test. I don´t think the curve representation is the best suitable for these result.

Authors can consider present results in two parts: ethanol in extracts is not detrimental for survival and on the other side, dependence of viability to 1,8 cineole concentration compared to non 1,8 cineole (first column on raw data).

Author response #3: As requested, the concentration-dependent data in the old Fig. 2A is now displayed in two parts: ethanol effect in one panel (A) and 1,8-cineole effect in other panel (B). We additionally performed an ANOVA test to analyze significance of differences in cell viability after treatment (see new Fig2).

Specific comment #4: Table 3. The dispersion values between assays are noteworthy. Although differences are

significant authors should address why they have such differences between assays. Dispersion was not observed in raw data for fig 1. I assume that some disruption of biofilm was produced during washes during treatment.

Author response #4: We have now included a sentence to discuss variations between assays (see L316-318).

Specific comment #5: Authors described viability of detached cells arguing “It has been postulated that a good

antibiofilm agent should not only attack bacteria into the biofilm but also display an action against biofilm-released cells”. The results is in concordance with high rate of dead cells on biofilm. It would be more interesting and will significantly improve the impact of the work to investigate if a post-treatment biofilm is able to growth.

Author response #5: We agree with the reviewer´s comment regarding post-treatment biofilms. Even though this experiment is beyond the scope of the present work, we now mention in the manuscript their importance (see L435-436).

Specific comment #6: Authors did not demonstrate that high biofilm biomasses yielded by strains are consequence of thicker extracellular matrix. Statement in L388-L393 should be modified.

Author response #6: The requested statement has been modified (see L415-416).

We thank the reviewers and editors for their constructive comments and suggestions. We believe that the revised manuscript is now acceptable for publication in PlosOne.

Sincerely yours,

Estela Galván

---

## [Decision Letter · Decision Letter 2]

12 Aug 2020

PONE-D-20-04063R2

Cell death and biomass reduction in biofilms of multidrug resistant extended spectrum β-lactamase-producing uropathogenic Escherichia coli isolates by 1,8-cineole

PLOS ONE

Dear Dr. Galvan,

Thank you for submitting your manuscript to PLOS ONE. After careful consideration, we feel that it has merit but does not fully meet PLOS ONE’s publication criteria as it currently stands. Therefore, we invite you to submit a revised version of the manuscript that addresses the points raised during the review process.

We look forward to receiving your revised manuscript.

Kind regards,

Monica Cartelle Gestal, PhD

Academic Editor

PLOS ONE

Reviewers' comments:

Reviewer's Responses to Questions

**Comments to the Author**

1. If the authors have adequately addressed your comments raised in a previous round of review and you feel that this manuscript is now acceptable for publication, you may indicate that here to bypass the “Comments to the Author” section, enter your conflict of interest statement in the “Confidential to Editor” section, and submit your "Accept" recommendation.

Reviewer #2: All comments have been addressed

Reviewer #3: (No Response)

2. Is the manuscript technically sound, and do the data support the conclusions?

Reviewer #2: Yes

Reviewer #3: Yes

3. Has the statistical analysis been performed appropriately and rigorously? 

Reviewer #2: Yes

Reviewer #3: Yes

4. Have the authors made all data underlying the findings in their manuscript fully available?

Reviewer #2: Yes

Reviewer #3: Yes

5. Is the manuscript presented in an intelligible fashion and written in standard English?

Reviewer #2: Yes

Reviewer #3: Yes

6. Review Comments to the Author

Reviewer #2: (No Response)

Reviewer #3: Although most of the comments have been addressed, there are still a number of flaws that need to be addressed.

# one

The authors cite an article by Kragh et al to justify the stochastic variation in the biofilm results for the 72-hour biofilm and discourage the use of the CV staining method to draw conclusions. However, variations in biomass quantification were observed by microscopy (CV results were not shown) and were not as large as presented in this manuscript.

Authors must discuss the comment more extensively than citing the article by Kragh et al.

# two

Specific comment # 5. I disagree with the authors. I think the suggested experiment would improve significantly

impact of work. The authors present a new drug capable of altering and killing the cells presented in the biofilm life form. It is important to determine if the cells can grow.

7. PLOS authors have the option to publish the peer review history of their article (what does this mean?). If published, this will include your full peer review and any attached files.

Reviewer #2: **Yes: **James Gurney

Reviewer #3: No

---

## [Author Response · Author response to Decision Letter 2]

24 Sep 2020

Estela M. Galvan, Ph D

Lab. of Bacterial Pathogenesis

Centro de Estudios Biomedicos, Biotecnologicos,

Ambientales y Diagnostico (CEBBAD)

Universidad Maimonides

Hidalgo 775

C1405BWE-Buenos Aires, Argentina

September 24, 2020

Dear Monica Cartelle Gestal, PhD

Academic Editor

PLOS ONE

Please find enclosed our revised version of the manuscript number PONE-D-20-04063R2 entitled “Cell death and biomass reduction in biofilms of multidrug resistant extended spectrum β-lactamase-producing uropathogenic Escherichia coli isolates by 1,8-cineole” for your consideration. It has been revised and changes have been made for each specific comment of the reviewers, as addressed below (modifications are indicated, referring to line numbers in the Marked Up Manuscript file).

Reviewer #3: 

General comment: Although most of the comments have been addressed, there are still a number of flaws that need to be addressed.

Concern #1: The authors cite an article by Kragh et al to justify the stochastic variation in the biofilm results for the 72-hour biofilm and discourage the use of the CV staining method to draw conclusions. However, variations in biomass quantification were observed by microscopy (CV results were not shown) and were not as large as presented in this manuscript. Authors must discuss the comment more extensively than citing the article by Kragh et al.

Author response #1: As requested, we have now discuss more extensively the experimental variables that could account for the large biomass variations observed among biological replicates in the stronger biofilm-producer strains, particularly in the reference strain Ec ATCC25922 (see L231-242).

Concern #2: Specific comment # 5. I disagree with the authors. I think the suggested experiment would improve significantly impact of work. The authors present a new drug capable of altering and killing the cells presented in the biofilm life form. It is important to determine if the cells can grow.

Author response #2: As requested, new experimental data related to whether a post-treatment biofilm is able to regrow have been added (see L41-42, L169-176, L391-409 –including new Table 4-, L453-458).

We thank the reviewers and editors for their constructive comments and suggestions. We believe that the revised manuscript is now acceptable for publication in PlosOne.

Sincerely yours,

Estela Galván

---

## [Decision Letter · Decision Letter 3]

26 Oct 2020

Cell death and biomass reduction in biofilms of multidrug resistant extended spectrum β-lactamase-producing uropathogenic Escherichia coli isolates by 1,8-cineole

PONE-D-20-04063R3

Dear Dr. Galvan,

We’re pleased to inform you that your manuscript has been judged scientifically suitable for publication and will be formally accepted for publication once it meets all outstanding technical requirements.

Kind regards,

Monica Cartelle Gestal, PhD

Academic Editor

PLOS ONE

Additional Editor Comments (optional):

Reviewers' comments:

Reviewer's Responses to Questions

**Comments to the Author**

1. If the authors have adequately addressed your comments raised in a previous round of review and you feel that this manuscript is now acceptable for publication, you may indicate that here to bypass the “Comments to the Author” section, enter your conflict of interest statement in the “Confidential to Editor” section, and submit your "Accept" recommendation.

Reviewer #2: All comments have been addressed

Reviewer #3: All comments have been addressed

2. Is the manuscript technically sound, and do the data support the conclusions?

Reviewer #2: Yes

Reviewer #3: (No Response)

3. Has the statistical analysis been performed appropriately and rigorously? 

Reviewer #2: Yes

Reviewer #3: (No Response)

4. Have the authors made all data underlying the findings in their manuscript fully available?

Reviewer #2: Yes

Reviewer #3: (No Response)

5. Is the manuscript presented in an intelligible fashion and written in standard English?

Reviewer #2: Yes

Reviewer #3: (No Response)

6. Review Comments to the Author

Reviewer #2: While I signed off on the last version of this manuscript, I agree with reviewer 3 that the requested additions have strengthened the paper. My old comment is that I think the regrowth control should not have been a culture that had reach carrying capacity. Instead, I would have preferred to see the authors disrupt and dilute the biofilm so that the CFU was close to the treated level and watch for regrowth. However given the clear results of lower regrowth, I think the current assay is sufficient.

James Gurney

Reviewer #3: (No Response)

7. PLOS authors have the option to publish the peer review history of their article (what does this mean?). If published, this will include your full peer review and any attached files.

Reviewer #2: **Yes: **James Gurney

Reviewer #3: No

---

## [Editor Report · Acceptance letter]

28 Oct 2020

PONE-D-20-04063R3 

Cell death and biomass reduction in biofilms of multidrug resistant extended spectrum β-lactamase-producing uropathogenic Escherichia coli isolates by 1,8-cineole 

Dear Dr. Galván:

I'm pleased to inform you that your manuscript has been deemed suitable for publication in PLOS ONE. Congratulations! Your manuscript is now with our production department. 

Kind regards, 

on behalf of

Dr. Monica Cartelle Gestal 

Academic Editor

PLOS ONE